# Meeting SDG6 in the Kingdom of Tonga: The Mismatch between National and Local Sustainable Development Planning for Water Supply

**Ian White [1,\*], Tony Falkland [2] and Taaniela Kula [3]**

[1]   Fenner School of Environment and Society, Australian National University, Canberra, ACT 0200, Australia

[2]   Island Hydrology Services, 9 Tivey Place, Hughes, Canberra, ACT 2605, Australia;
tony.falkland@netspeed.com.au

[3]   Natural Resources Division, Ministry of Lands and Natural Resources, Nuku'alofa, Tonga;
tkula@naturalresources.gov.to

\*   Correspondence: ian.white@anu.edu.au; Tel.: +61-418-262-881

**Abstract:** UN Sustainable Development Goal 6 challenges small island developing states such as the Kingdom of Tonga, which relies on variable rainwater and fragile groundwater lenses for freshwater supply. Meeting water needs in dispersed small islands under changeable climate and frequent extreme events is difficult. Improved governance is central to better water management. Integrated national sustainable development plans have been promulgated as a necessary improvement, but their relevance to island countries has been questioned. Tonga's national planning instrument is the Tonga Strategic Development Framework, 2015–2025 (TSDFII). Local Community Development Plans (CDPs), developed by rural villages throughout Tonga's five Island Divisions, are also available. Analyses are presented of island water sources from available census and limited hydrological data, and of the water supply priorities in TSDFII and in 117 accessible village CDPs. Census and hydrological data showed large water supply differences between islands. Nationally, TDSFII did not identify water supply as a priority. In CDPs, 84% of villages across all Island Divisions ranked water supply as a priority. Reasons for the mismatch are advanced. It is recommended that improved governance in water in Pacific Island countries should build on available census and hydrological data and increased investment in local island planning processes.

**Keywords:** groundwater; rainwater harvesting; climate variability; small island developing states; improved water governance; national sustainable development plans; SDG6; community participation

## 1. Introduction

The United Nations (UN) 2030 Sustainable Development Goal (2030) for water and sanitation, i.e., "Ensure availability and sustainable management of water and sanitation for all" (SDG 6 [1]), presents significant challenges for small island development states (SIDS), particularly in terms of securing universal and equitable access to safe and affordable water for all. Limited resources and institutions, dispersed island communities, increasing demands, scarce fresh groundwater resources vulnerable to salinization and pollution, variable and changing climates driven by large-scale ocean-atmosphere interactions and frequent, devastating, extreme events such as tropical cyclones, droughts and floods compound the difficulties of ensuring that communities have access to adequate and safe freshwater supplies [2,3], which is fundamental for sustainable development.

Faced with these difficulties, Oceania was one of the few regions in the world that did not meet the 2015 Millennium Development Goals for water and sanitation [4]. Better governance and increased water information have been claimed to be key factors in improving water security in Pacific

island countries (PICs), particularly when facing impacts of climate change [5,6]. National strategic development strategies (NSDS) for SIDS have been promoted as a key mechanism for improved governance and for fulfilling government commitments to local, regional and international agendas on sustainable development [7], particularly SDGs, the 2005 Mauritius Strategy for the Further Implementation of the Programme of Action for the Sustainable Development of SIDS [8] and the 2014 SIDS Accelerated Modalities of Action (SAMOA) Pathway resolution [6].

NSDSs are used to identify national priorities, to allocate resources to government agencies and to monitor outcomes. Initially, creating an NSDS involved a two-phase approach, with national assessments in phase one, followed by selected interventions in phase two [6]. Inexperience, limited resources and institutions in some SIDS meant that donor and funding agency-supported external consultants played central roles in driving NSDS processes assisted by senior national bureaucrats. These planning processes tend to be top-down prescriptions, with the planning priorities and expected outcomes being predominantly economically focused. The applicability of similar, transplanted governance mechanisms to PICs has been questioned [9].

One of the key characteristics of SIDS, and particularly those with dispersed island communities, is their well-developed local institutions which are particularly suited to bottom-up, priority-setting processes [10]. Their advantage is inclusiveness, but a draw-back of bottom-up processes is the time involved to reach agreement or consensus [11]. Planners are faced with a dilemma: are efficient top-down national planning processes able to address the priorities of local communities, or are lengthy, expensive community processes required? The South Pacific Kingdom of Tonga, reliant predominantly on fresh groundwater lenses and rainwater harvesting for water supply, presents a unique opportunity to compare priorities in water supply identified by both top-down and bottom-up planning processes.

The government of Tonga, with support from the Asian Development Bank, developed the Tonga Strategic Development Framework 2015–2015 (TSDFII) through a high-level consultation process over three months in late 2014 [12]. TSDFII incorporated lessons learnt in the preceding Tonga Strategic Development Framework 2011–2014. TSDFII identified 29 highest priority issues as key planned "Organisational Outcomes" (OOs) for Ministries, Departments and agencies, while 153 more issues raised during consultations are listed as strategic concepts (SCs) under the OOs. These are meant to be planning aids to inform Ministry corporate plans and decisions regarding budgeting, staffing and reports.

From 2007 to 2016, a very extensive, nation-wide Community Development Plan (CDP) process, supported by a range of agencies, involved most rural villages throughout Tonga's five Island Divisions. Communities identified and ranked priority development issues in their village and then built and endorsed their CDP. The plans prioritized the most urgent issues in each village in terms of women's, youths' and men's perspectives, and 136 CDPs were presented in 2016 [13].

One of the difficulties in assessing water security and priorities in PICs is the limited information on water availability and its use, particularly in rural areas [3]. Census data usually provides relatively coarse information on household water sources. In this work, four questions are addressed:

1. In the absence of detailed water data, can census data, together with available hydrological data, be used to asses planning priorities for water supply in dispersed island countries?
2. Do variations in rainfall due to climate change need to be addressed for water supply in medium-term (10 years) national development plans?
3. Do top-down governance templates, such as national sustainable development strategies, capture the priorities of rural island communities in water supply?
4. Can national development plans be improved for water supply in multi-island countries?

In this work, we summarize census demographic and water source data, and limited information on water use, groundwater and rainfall characteristics of the Kingdom of Tonga, to identify Island Divisions with special needs. We then analyze the priority given to water supply in the national

planning instrument, TSDFII. This is compared with Island Division-level priorities found by analyzing water supply priorities in the 117 available CDPs. These are analyzed in terms of women's, youths' and men's priorities. Island Division priorities are related to the analysis of census and hydrological data. The bottom-up CDP priorities in water supply are contrasted with those in the top-down TSDFII. The discrepancy between national and local priorities in water supply are discussed, and suggestions for improving processes are made.

## 2. Materials and Methods

### 2.1. Study Location

The Kingdom of Tonga's population of 102,000 people is dispersed over 169 islands in five Island Divisions covering a land area of about 750 km$^2$ scattered across 700,000 km$^2$ of the southwestern Pacific Ocean (Figure 1). The Kingdom adjoins the seismically very-active Tonga trench. Tonga's western islands are of volcanic origin, while the eastern islands are uplifted coral limestone and sand islands. Many of the eastern islands, such as Tongatapu, the largest island which contains the capital Nuku'alofa, have a mantle of fertile volcanic soil from past volcanic eruptions of the western islands. Volcanic eruptions, earthquakes and subsequent tsunamis, tropical cyclones (TCs), storm surges and El Niño Southern Oscillation- (ENSO) and Pacific Decadal Oscillation (PDO)-related droughts and floods are frequent natural hazards faced by the Kingdom's island communities. Recent TCs that have devastated parts of Tonga include TC Ian (2014), TC Gita (2018) and TC Harold (2020).

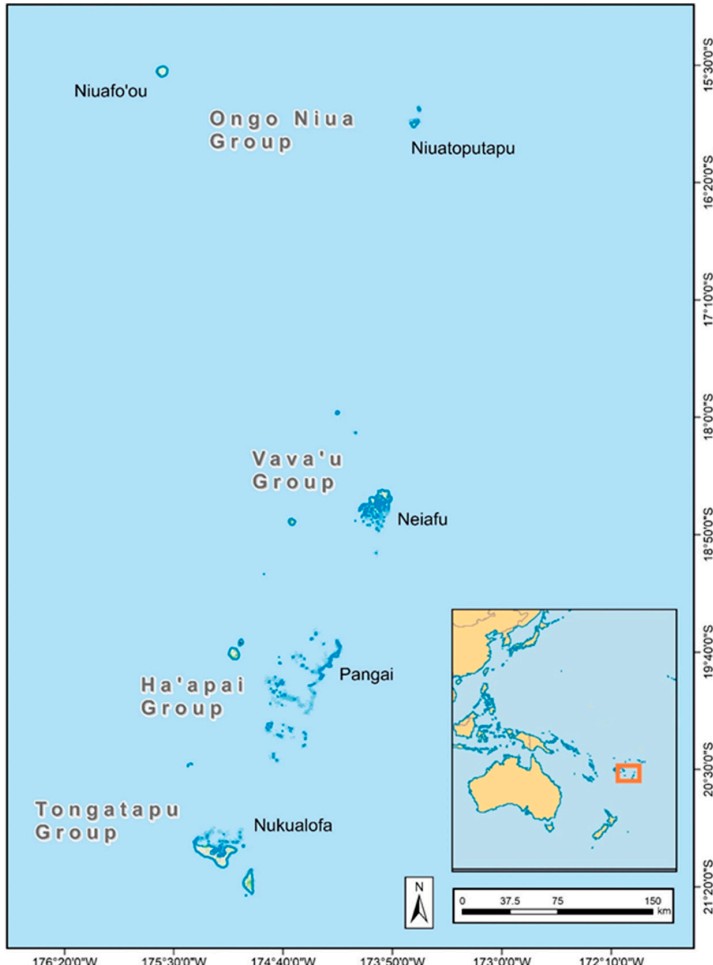

**Figure 1.** Map of the Kingdom of Tonga, showing main island Divisions and population centers [14].

Annual rainfall in Tonga increases from south to north, influenced by proximity to the South Pacific Convergence Zone (SPCZ). Annual rainfall variability is moderate and smaller in the north than in the south. Tonga has a wetter season from November to April, followed by a drier season from May to October, also influenced by the position of the SPCZ. The predominant sources of water are rainwater harvesting, which Tongans prefer for drinking, and groundwater from groundwater lenses or springs in the carbonate and sand islands, used for all other purposes including washing, bathing and sanitation. Groundwater salinity varies from fresh to brackish and even saline, depending on island geology, size, and ENSO and PDO conditions. In general, many of the volcanic islands do not appear to have useable groundwater [14].

In 2016, nearly three quarters of Tongans lived on the main island Tongatapu, with Greater Nuku'alofa, the capital region, having over a third of the country's population. Just over a quarter of the population are spread over the Kingdom's other four Island Divisions. About 55% of the population are under the age of 25, with youths aged 14 to 24 making up nearly 19% of the total population. Gross domestic product (GDP) in 2017 was estimated to be US$5900 per person, with an annual growth rate around 2.5% [15]. Per capita GDP varies widely between island groups, with Tongatapu being 15% above the national average in 2013, while Ha'apai was 40% below the national average. Natural disasters have been estimated to cost on average 4.4% of GDP and TC Ian in 2014, which affected 70% of the population of Ha'apai, cost 11% of GDP [10].

Community piped water supplies are sourced from groundwater, springs or, less commonly, from community rain tanks [14]. In Nuku'alofa, 'Eua and urban centers on Vava'u and Ha'apai, piped water is supplied by the Tonga Water Board. In villages throughout Tonga, piped water supplies are the responsibility of Village Water Committees, overseen by the Ministry of Health. Piped water supply in villages is often intermittent.

## 2.2. Demographics, Water Sources, Water Use and Water Demand

We use the results of the latest census in 2016 [16] to examine the urban/rural composition and the distribution of population and their trends across Tonga's five Island Divisions (Figure 1). Census results are also used to compare water sources used by households for different water uses, again between urban and rural communities and across Island Divisions. As a measure of the ease of accessing water supply for the greatest volumetric use of water for nondrinking purposes [17], we use the water source ratio (WSR), defined for the number of households (HH) at the Island Division level as:

$$WSR = (HH \text{ with rain water supply})/(HH \text{ with piped water supply}) \tag{1}$$

Outside urban areas and population centres, there is very little available information on water demand and unaccounted for water in Tonga. Here, we make use of recent estimates of water supply, $Q$ (ML/day), for urban areas [14], and the small number of estimates of daily per capital water use, $W$ (L/pers/day), for a handful of villages in outer islands [18–20]. For the urban areas, $W$ was estimated from:

$$W = 10^6 \times Q \times (1 - UAW/100)/N, \tag{2}$$

where UAW is the percentage of unaccounted for water, and $N$ the number of urban inhabitatnts.

## 2.3. Climate, Rainfall, Variability and Rainwater Harvesting Failures

A recent draft report on national water resources in Tonga [14] is used to extract data on average annual and seasonal rainfall and their variability and trends across Tonga's Island Divisions. Trends in annual rainfall are examined using linear regression and their statistical significance is tested. Trends are compared with climate model projections of the impact of climate change on rainfall in Tonga [21].

The frequencies of failure of rainwater harvesting systems over the six month wet (Novemeber to April) and dry (May to October) seasons are estimated from the percentile of 360 mm of rainfall over

the full seasonal record for that location. The daily per capita water availability, S (L/person/day), for a household rainwater harvesting is given by:

$$S = CE \times P_S \times A/(d_S \times N_{HH}) \tag{3}$$

where $P_s$ (mm) is the seasonal rainfall, CE is the capture efficiency of rainwater harvesting, A ($m^2$) is the roof catchment area, $d_S$ is the number of days in the season and $N_{HH}$ is the number of people in a household.

When $P_S = 360$ mm, and with a typical roof area of 100 $m^2$, a capture efficiency of 0.55 [18–20] and the average household size in Tonga (5.5 persons), Equation (3) gives S of less than 20 L/person/day, i.e., below the World Health Organization recommended minimum quantity of water required to satisfy essential health and hygiene needs in emergency situations [17].

*2.4. Groundwater*

A summary of the limited available information on groundwater resources is provided in [14], and the few available village integrated water management plans in [18–20]. Other information is sourced from reports and publications [22–28]. Estimates of per capita groundwater use in population centers are based on water supply data provided by the Tonga Water Board [29].

*2.5. Tonga Strategic Development Framework 2025–2015*

The TSDF II is: "the overarching framework of the planning system in Tonga. It provides an integrated vision of the direction that Tonga seeks to pursue." [12]. TSDFII is conceived as the top of a cascading system of planning and budgeting in Tonga which is intended to guide:

- medium-term sector and district/island master plans
- three year rolling Corporate Plans and Budgets for all Ministries, Departments and Agencies
- annual Divisional and Staff Plans and job descriptions.
- consultation, monitoring, and evaluation.

TSDF II identifies Government priorities, assigns Ministerial responsibilities and aims to focus resources. It is arranged in a hierarchy where 29 Organizational Outcomes (OO), grouped under three institutional pillars and two input pillars, feed into seven desired National Outcomes (NOs) which, in turn, feed into the single planned National Impact of the TSDF II: "A more progressive Tonga supporting a higher quality of life for all", which supports the Motto of TSDF II, given by the reformer monarch Tupou I: "God and Tonga are my inheritance." TSDFII also lists 153 Strategic Concepts (SCs) which were issues raise during the consultation process which lie outside TDSFII, but are intended as aids to sector, district and Ministry, Department and Agency planning and budgeting.

The Ministry of Finance and National Planning (MFNP) led the creation of TSDF II, which was based on a wide, but fairly rapid consultative process. In the period from October 2014 to December 2014, consultation meetings were held throughout Tongatapu and the Island Divisions of 'Eua, Ha'apai and Vava'u. The Ongo Niua Division was not visited.

The TDSFII was scanned for references to water, freshwater, rainwater, groundwater and water supply, and the planned National Outcomes, OOs and SCs were examined for their relevance to water supply. In NSDS, water supply is usually identified as an infrastructure service. The weight given to water supply in TDSFII is assessed relative to the weight given to infrastructure and other services identified in TSDFII, such as energy, transport, information and communications technology, building and structures, and research and development, in the listed OOs and SCs.

*2.6. Community Development Plans, 2016*

Work on CDPs began in 2007, under the Local Government Division of the Ministry of Internal Affairs. The CPDs were a response to the then National Vision, i.e., "a Progressive Tonga Supporting

Higher Life for All." Consultations throughout rural villages in Tonga's five Island Divisions were implemented by the nongovernment organization, Mainstreaming of Rural Development Innovation Trust Tonga (MORDI TT). The CDP process was supported by the International Fund for Agricultural Development, United Nations Development Programme, Australian Agency for International Development and the Tonga Government. One of the requirements of the project was the participation of 80% of the population of each rural village in the development, ranking of priorities and endorsement of the village CDP. This was a lengthy process which culminated in District Officers and Town Officers of 136 village communities presenting their CDPs to the then Prime Minister, the late Hon. 'Akilisi Pohiva on 4 October, 2016 [13].

During the planning process, the Department for Local Government was transferred from the Prime Minister's Office to the Ministry of Training, Employment, Youth, and Sports, and then to the Ministry of Internal Affairs. Analysis of and action on the Community Development Plans appears to have been deferred by these moves. We have been unable to find any analysis of the valuable information on island priorities contained in these CDPs.

Of the 136 CDPs presented, 117 are available online [30]. These represent over 77% of all rural villages in Tonga. CDPs were downloaded and the priority rankings of each village that mentioned water or water supply were examined and their ranking recorded (Table S1). Particular note was made when water or water supply ranked as the top priority or in the top three priorities for women, men and youths separately. Village level results were aggregated to Island Division level, and the percentage of villages in each Island Division that ranked water as the highest priority, that ranked it within the top three priorities or that ranked it anywhere within their list of priorities were recorded. A comparison was then made of the weight given to water supply as an infrastructure service in TSDFII and to indications of Island Division-level water supply limitations from the 2016 census and hydrological data.

## 3. Results

### 3.1. Demographics

The distribution of Tonga's population across its island Divisions in 2016 is given in Table 1 [16]. Also shown are the total rural, urban and the population of Greater Nuku'alofa, as well as the annual growth rate between 2011 and 2016, the population density and the average household size.

**Table 1.** Summary of 2016 population statistics for Tonga as a whole, for island Divisions, for urban, rural areas and for greater Nuku'alofa populations [16].

| Item | TONGA | 'Eua | Tongatapu | Vava'u | Ha'apai | Ongo Niua [1] | Urban [2] | Rural [3] | Greater [4] Nuku'alofa |
|---|---|---|---|---|---|---|---|---|---|
| Total Population | 100,651 | 4945 | 74,611 | 13,738 | 6125 | 1232 | 23,221 | 77,430 | 35,184 |
| Male | 50,255 | 2486 | 37,135 | 6866 | 3118 | 650 | 11,529 | 38,726 | 17,490 |
| Female | 50,396 | 2459 | 37,476 | 6872 | 3007 | 582 | 11,692 | 38,704 | 17,694 |
| Population change 2011–2016 (%) | −2.5 | −1.4 | −1.1 | −7.9 | −7.4 | −3.9 | −4.2 | −2 | −2.4 |
| Av. Annual Growth (%) | −0.5 | −0.3 | −0.2 | −1.7 | −1.5 | −0.8 | −0.9 | −0.4 | −0.5 |
| Population Density (pers/km$^2$) | 155 | 57 | 286 | 114 | 56 | 17 | 2035 | 121 | 1010 |
| Number of Households | 18,198 | 889 | 13,096 | 2745 | 1193 | 275 | 4175 | 14,023 | 6240 |
| Average Household Size (pers) | 5.5 | 5.6 | 5.7 | 5 | 5.2 | 4.5 | 5.6 | 5.5 | 5.6 |
| Number of Villages | 165 | 15 | 67 | 44 | 27 | 12 | 3 | 162 | 14 |

[1] Niuafo'ou and Niuatoputapu combined. [2] Urban area comprises the villages of Kolofo'ou, Kolomotu'a and Ma'ufanga in Tongatapu. [3] Rural area consists of all villages in Tonga except Kolofo'ou, Kolomotu'a and Ma'ufanga in Tongatapu. [4] Greater Nuku'alofa is made up of the districts of Kolofo'ou and Kolomotu'a in Tongatapu.

The demographic data in Table 1 show a concentration of population in the main island, Tongatapu, with 35% of the population of just over 100,000 living in Greater Nuku'alofa, and a further 40% of the population living in 53 rural villages across Tongatapu Island Division. Just over 35% of all rural villages are in Tongatapu. The remaining 26% of Tonga's population is scattered over 98 villages in Tonga's other four Island Divisions.

Between 2011 and 2016, the population of Tonga shrank by an annual rate of 0.5%, and urban populations decreased at over twice the annual rate of rural populations. The rate of loss of population was highest in Vava'u and Ha'apai Island Divisions and lowest in Tongatapu. In 2015, 70% of the housing stock in Ha'apai was devasted by TC Ian [12]. Table 1 suggests an inward migration from outer islands, particularly to rural villages in Tongatapu, at an annual rate of about +0.9%. This suggests that water demand due to the number of people should be decreasing in outer islands and increasing in Tongatapu.

Population density was over 16 times higher in urban areas that in rural areas. In Tongatapu, the population density was nearly 17 times higher than that of the far northern Ongo Niua Division. Average household size in urban areas, however, was very close to that in rural areas, with Tongatapu having the highest average household size (5.7 persons) and Ongo Niua the lowest (4.5 persons).

*3.2. Water Sources*

Table 2 lists the percentage of households using water from different sources for drinking and for all other uses for Tonga as a whole and for each of the island Divisions from the 2016 census [16].

**Table 2.** Percentages of households in Tonga as a whole, in each island Division, in urban and rural areas and Greater Nuku'alofa using water from different sources for (a) drinking water and (b) all other water uses given by the 2016 census [16].

| Water Source | TONGA | 'Eua | Tongatapu | Vava'u | Ha'apai | Ongo Niua | Urban | Rural | Greater Nuku'alofa |
|---|---|---|---|---|---|---|---|---|---|
| **(a) Drinking Water (%)** | | | | | | | | | |
| Piped Supply | 10.0 | 2.5 | 11.3 | 10.3 | 3.0 | 1.1 | 12.2 | 9.3 | 11.7 |
| Rain Tank | 60.5 | 73.1 | 54.8 | 69.2 | 86.2 | 92.3 | 53.5 | 62.5 | 53.8 |
| Neighbour/Community [1] | 19.6 | 23.4 | 20.7 | 18.8 | 10.0 | 6.6 | 14.3 | 21.2 | 15.8 |
| Bottled water | 9.5 | 0.9 | 12.8 | 1.4 | 0.6 | 0.0 | 19.3 | 6.6 | 18.1 |
| Boiled Well Water | 0.2 | 0.1 | 0.2 | 0.1 | 0.1 | 0.0 | 0.4 | 0.1 | 0.3 |
| **Other** | 0.2 | 0.0 | 0.2 | 0.2 | 0.2 | 0.0 | 0.3 | 0.1 | 0.2 |
| **(b) All Other Water Uses (%)** | | | | | | | | | |
| Piped Supply | 88.3 | 95.6 | 93.3 | 80.7 | 52.0 | 59.0 | 92.3 | 87.1 | 92.0 |
| Rain Tank | 10.9 | 4.0 | 6.0 | 18.9 | 44.6 | 40.3 | 7.1 | 12.0 | 7.1 |
| Own Well | 0.6 | 0.0 | 0.5 | 0.0 | 3.1 | 0.4 | 0.4 | 0.6 | 0.6 |
| Other | 0.3 | 0.5 | 0.3 | 0.3 | 0.3 | 0.4 | 0.2 | 0.3 | 0.3 |
| Total Number Households | 18,005 | 885 | 12,953 | 2715 | 1179 | 273 | 4089 | 13,916 | 6139 |

[1] Probably neighbour or community rain tanks.

Table 2 reveals the variety of water sources used for different purposes in Island Divisions. Households throughout Tonga prefer rainwater (over 60%) for drinking over piped supply (10%). Most piped water supply is sourced from groundwater. Nationally, nearly 20% of drinking water is sourced from neighbors' or community rain tanks.

Urban households in Table 2a have a 31% higher use of piped water for drinking than rural users, who access rainwater for drinking 17% more than urban users. Bottled water is increasingly being used for drinking, with urban use of bottled water being four times that of rural households. Boiled groundwater from household wells has very low use for drinking, i.e., 0.2% nationally. Local groundwater has the potential to be polluted from household sanitation systems, such as pit latrines or leaking septic tanks. Ha'apai and Ongo Niua stand out in the island Districts, with much less drinking water supplied from piped water than the other island Divisions, and correspondingly, much more from rainwater harvesting.

All other uses of water, i.e., bathing, washing, sanitation, make up the bulk of volumetric demand for water [17]. When all other uses of water are considered, Table 2b shows that nationally, 88% of household water is sourced from piped groundwater systems, with rain tanks supplying only 11% of other uses, since it is reserved for drinking. Only 0.6% access water from private wells. Rural households access piped water 5.7% less than urban households for nondrinking water uses;

therefore, rural households rely on rain tanks nearly 1.7 times more than those in urban districts. Again, Ha'apai and Ongo Niua are quite distinct from the other Island Divisions, with less than 60% of other household water use coming from the piped system and more than 40% coming from rainwater harvesting. Figure 2 shows the water source ratio, Equation (1) for other uses in each Island Division and Tonga as a whole. There is a large difference in WSR between, Ha'apai, Ongo Niua, Vava'u and Tongatapu and 'Eua. Piped water systems are less available in the Ha'apai and Ongo Niua Divisions than in the other Island Divisions [14]. This suggests that access to reliable water sources is more difficult in those islands.

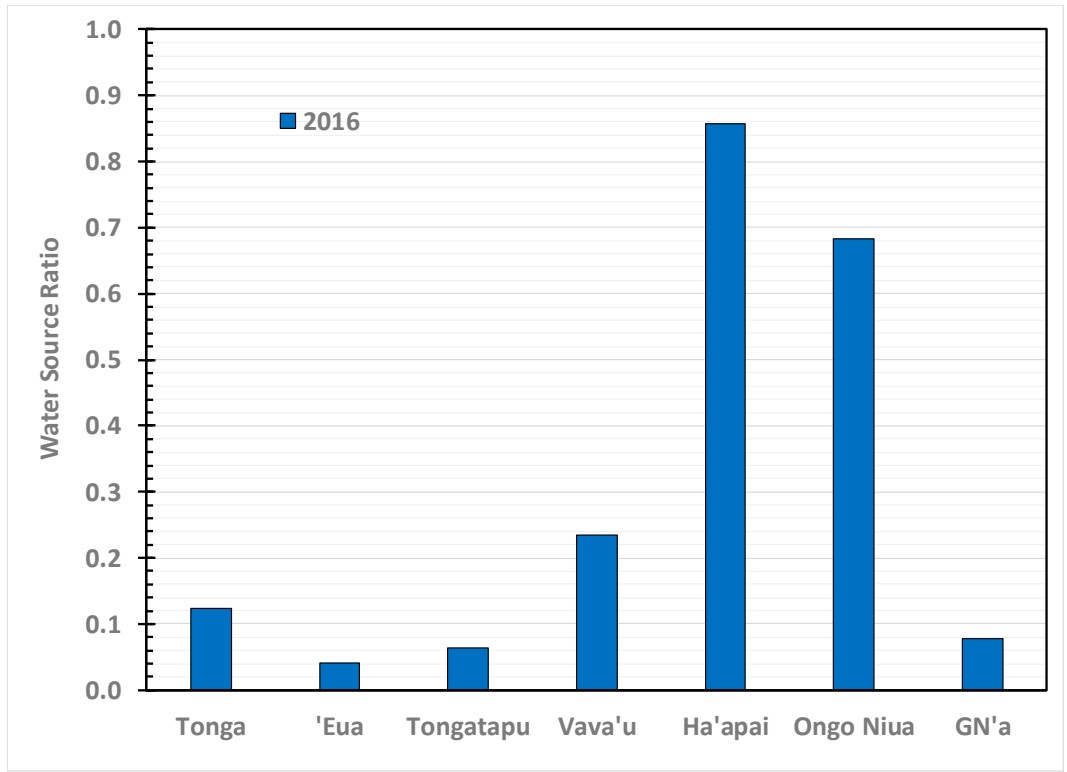

**Figure 2.** Average water source ratio for nondrinking uses in households for Tonga as a whole, for Tonga's Island Division, and for Greater Nuku'alofa (GN'a).

*3.3. Fresh Groundwater*

Fresh groundwater in Tonga's islands mostly consists of freshwater lenses overlying seawater in carbonate (limestone and sand) islands [14,22–26]. The salinity gradient increases with depth through the lens, going from low salinity water at the groundwater surface through a diffuse saline transition zone to underlying seawater [24]. Thin freshwater lenses have higher salinity than thicker lenses, but salinity varies with rainfall recharge, pumping and island geomorphology [5]. Groundwater may also exist in fractured rock aquifers in the volcanic islands, but these are yet to be assessed [14]. There are also a number of springs emanating from fresh perched groundwater on 'Eua, an island with mixed volcanic and carbonate geology, and small freshwater lakes on Niuafo'ou [14].

There is useful information about the freshwater lens in Tongatapu, used to supply the capital Nuku'alofa from a nearby wellfield location at Mataki'eua-Tongamai, but less about village pumping from local wells and boreholes throughout Tongatapu, in [22–25,27,28,31]. Local wells and boreholes are vulnerable to contamination from leaking septic tanks and pit latrines.

The mean maximum height of the surface of the freshwater lens in Tongatapu has been estimated to be about 0.6 m above mean sea level, and the lens thins toward the coastal margins. Its elevation varies slightly with the oceanic tide. In the period from 1997 to 2018, the maximum thickness of the

freshwater lens was around 16 m. Maximum thickness varied with rainfall between 9.5 and 16 m. During the same period, the salinity of water, as measured by the electrical conductivity (EC) of the water supplied to Nuku'alofa, varied between almost 1600 µS/cm, following the dry period in 1998, to about 800 µS/cm, following the wet period in 1999 [14].

There is no evidence of any increasing trend in groundwater salinity in Tongatapu over the period from 1997 to 2018 [14]. Increases in EC of individual bores due to progressive increases in wellfield pumping rates over time have been observed at Makahi'eua-Tongamai since 1971 (Figure 3 [27]). The current rate of fresh groundwater extraction across all Tongatapu has been estimated to be about 24,000 m$^3$/day [29].

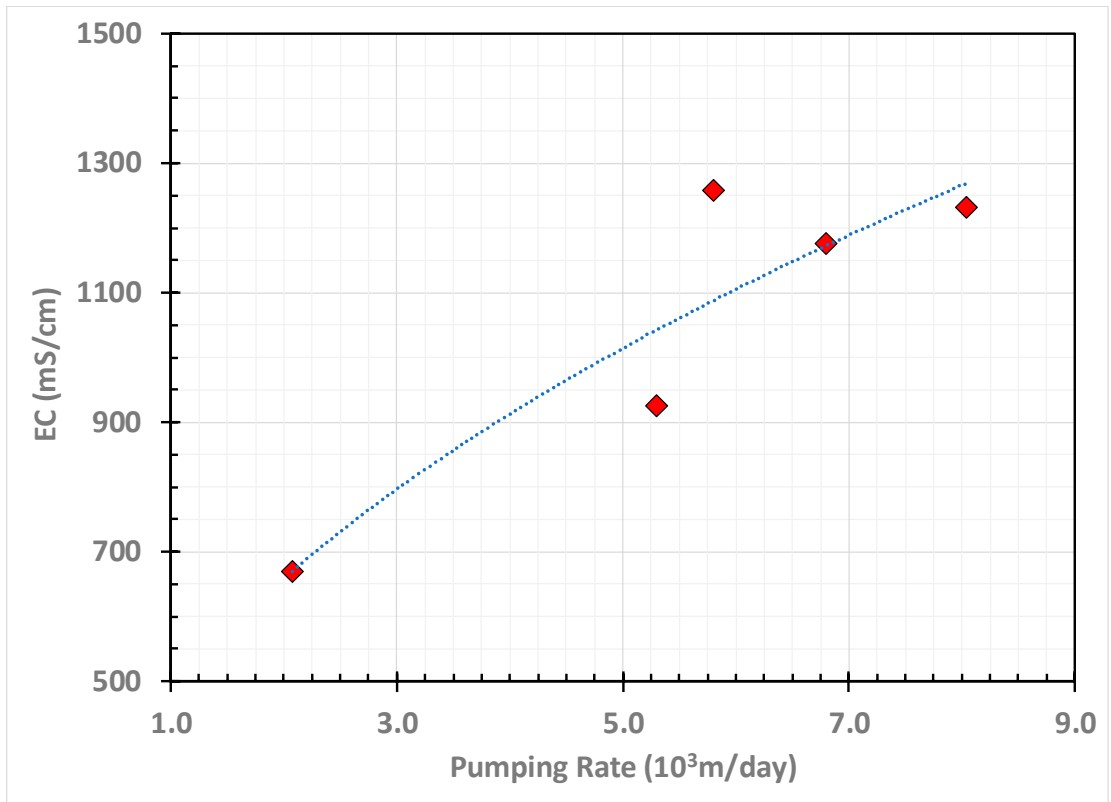

**Figure 3.** Impact of pumping rate from the Mataki'eua-Tongamai wellfield on the EC of water pumping from an individual well (PS106) between 1971 and 2007 [27].

There is limited information about groundwater used to supply population centers in Ha'apai or Vava'u. On Lifuka Island in the Ha'apai Division, the spatially limited fresh groundwater lens is much thinner than in Tongatapu, with maximum thickness ranging from 2 m in 1998 to 4 m at the end of 2011. Because of the thin freshwater lens, the salinity in the water supply system is higher than in Nuku'alofa, varying from a brackish EC of about 10,000 µS/cm to a low of around 1000 µS/cm. The salinity of the water there depends on the method of extraction, with groundwater pumped from vertical wells having a higher salinity, due to upconing of the underlying seawater, than that pumped from horizontal infiltration galleries or skimming wells. The approximate groundwater extraction rate for water supply in Lifuka is a about 500 m$^3$/day [29].

The population center of Vava'u, Neiafu, in the main island Vava'u 'Utu, a raised limestone island, is supplied water from a fresh groundwater lens with maximum thickness between 5 to 8.5 m over the period from 1999 to 2018. The thickness of the lens varies across the island, with some village wells only producing brackish groundwater. The salinity of water supply to Neiafu in this period

varied, from an EC of about 500 μS/cm in September 2000 to a brackish 4500 μS/cm in January 2016. The approximate rate of groundwater extraction to supply Neiafu is about 1000 m³/day [29].

In 'Eua, water is supplied from groundwater springs and wells. Salinity of the supply water in 'Eua is low, typically less than 500 μS/cm, and the supply of water is about 1100 m³/day [24,27].

### 3.4. Water Demand

With the diversity of water sources used (Table 2) and the differences in use between rural and urban and Island Divisions, there is little information on the ranges of water demand throughout Tonga. For Greater Nuku'alofa, about 12.6 ML/day are pumped from the groundwater Mataki'eua-Tongamai wellfield [13]. Unaccounted for water amounts to between 30 to 40% [29]. It has been estimated that about 11 ML/day of groundwater is extracted to supply the rural villages throughout Tongatapu [28]. Information is also available on groundwater extraction for the urban centers in Vava'u and Ha'apai [29]. There is limited information on unaccounted for water in these or in Tongatapu's village water supply systems. Inspections of village groundwater supply systems suggest that 40 to 50% losses are conservative. With this estimate and the population numbers in the 2016 census [16], the estimated average per capita groundwater use, Equation (2), in Greater Nuku'alofa, in rural villages in Tongatapu and in the urban centers in Vava'u and Ha'apai, as well as spring and groundwater demand in 'Eua, are given in Table 3.

**Table 3.** Estimated average per capita water use, Equation (2), in population centers and measured use in some rural villages in Tonga's Island Divisions supplied by spring water (SW), groundwater(GW) and rainwater (RW) [14,18–20,29,31].

| Island Division | Location | Water Source | Water Supply Rate ($10^3$ m³/Day) | Unaccounted for Water (%) | Average Water Use (L/Pers/Day) |
|---|---|---|---|---|---|
| 'Eua | | SW, GW | 1.1 | 40–50 | 110–130 |
| Tongatapu | Greater Nuku'alofa | GW | 12.6 | 30–40 | 230–270 |
| | Rural [1] Villages | GW | 11 | 40–50 | 130–160 |
| Vava'u | Neiafu | GW | 1.0 | 40–50 | 140–160 |
| | Koloa | RW | - | - | 18 |
| Ha'apai | Lifuka | GW | 0.5 | 40–50 | 125–150 |
| | Nomuka | RW | - | - | 22 |
| Ongo Niua | Niuafo'ou | RW | - | - | 14 |

[1] All villages not within Greater Nuku'alofa.

Rainwater is also used extensively throughout Tonga (Table 2). There is very little information on water use from rainwater harvesting systems. Measurements at villages on three outer islands, i.e., Koloa, Nomuka and Niuafo'ou, in the Vava'u, Ha'apai and Ongo Niua Divisions provide details on the average per capita rainwater and are listed in Table 3 [18–20]. In Koloa, the thin groundwater lens is saline and is only used for toilet flushing, cleaning and bathing [18]. In Nomuka, the village piped water supply, which was sourced from beneath a shallow ephemeral freshwater lake, is no longer operational, so household and community rain tanks are the only sources of freshwater [19].

Niuafo'ou island is a basalt shield volcano surmounted by an andesitic cone. Communities there rely on rainwater harvesting. It is not known if there is any viable fresh groundwater in Niuafo'ou [19]. The World Health Organization recommends that the minimum quantity of water required to satisfy essential health and hygiene needs in emergency situations is 20 L/person/day [17], a value similar to the regular average rainwater use for most purposes in these three outer islands.

## 3.5. Rainfall

Tonga's climate is tropical, with a warm period from December to April, when temperatures can reach 32 °C, and a cooler season from May to November, with temperatures infrequently rising above 27 °C [32]. Tonga's reliance of water sourced from rainfall harvesting and from groundwater (Table 2) means that rainfall and subsequent groundwater recharge are key determinants of water availability [22, 24,27,28]. Table 4 summarizes the rainfall characteristics at the six long-term meteorological stations throughout Tonga.

**Table 4.** Annual, wet (November to April) and dry season (May to October) rainfall characteristics at meteorology stations in Tonga [14].

| Met Station | Island Division | Daily Average Temperature Range (°C) [1] | Average Annual Rainfall (mm) | CV of Annual Rainfall | Mean Nov–Apr Rainfall (mm) | Mean May–Oct Rainfall (mm) | Period |
|---|---|---|---|---|---|---|---|
| Niuafo'ou | Ongo Niua | 25.9–27.9 | 2534 | 0.22 | 886 | 1648 | 1971–2019 |
| Niuatoputapu | | 25.7–27.5 | 2315 | 0.21 | 803 | 1512 | 1947–2019 |
| Lupepau'u | Vava'u | 22.9–26.9 | 2290 | 0.22 | 793 | 1497 | 1947–2019 |
| Ha'apai | Ha'apai | 23.1–27.4 | 1754 | 0.24 | 599 | 1155 | 1947–2019 |
| Fua'amotu | Tongatapu | 21.4–26.6 | 1765 | 0.25 | 664 | 1101 | 1980–2019 |
| Nuku'alofa | | 21.8–27.2 | 1863 | 0.26 | 735 | 1128 | 1945–2019 |

[1] Temperatures shown for the period from 1980 to 2017.

Average annual rainfall varies from about 1750 mm in the south to about 2500 mm in the northern islands closer to the Equator. Tonga has a wetter season from November to April and a drier season from May to October. Its rainfall is influenced by the position and strength of the South Pacific Convergence Zone (SPCZ) which is influenced both by season, ENSO events [21,33] and by the Pacific Decadal Oscillation (PDO) [24]. Tonga is periodically impacted by TCs whose intensity appears to be increasing [34], in line with climate change projections [21,35].

During the wet season, the SPCZ lies between Samoa and Tonga, while during the dry season, the SPCZ is normally positioned to the north-east of Samoa, where it is often weak, inactive and sometimes nonexistent [32]. In the northern islands, about 35% of annual rainfall occurs in the November to April period, while in the south, the percentage is slightly higher, i.e., 38–39%.

Estimates of average annual potential evaporation in Tonga range from about 1460 mm in Tongatapu to about 1670 mm in Niuatoputapu [32]. These high annual potential evaporation rates mean that open water storages lose large volumes of water. The losses from groundwater systems due to evaporation are much lower; therefore, groundwater storage has an advantage over surface storage.

## 3.6. Variability of Rainfall

The coefficient of variability (CV) of annual rainfall in Table 4 is moderate, at around 0.21 to 0.26, and is less in the wetter, northern islands than in the south. ENSO and the Pacific Decadal Oscillation (PDO) are key drivers of this variability. Two indicators of ENSO, the Niño Index (sea surface temperature anomaly in the Niño 3.4 region) and the Southern Oscillation Index (based on sea level pressure difference between Darwin and Tahiti), are strongly negatively and positively correlated, respectively, to wet season but not dry season rainfall. The PDO index is also strongly negatively correlated to wet season rainfall. In La Niña phases of ENSO (negative Niño Index), the SPCZ tends to move further south, and Tonga gets more rain in the wet season, while in El Niño phases (positive Niño Index), it moves further north and causes lower wet season rainfall [21]. Droughts tend to occur during El Niño events. During severe El Niños, the SPCZ can spread azonally along the equator [36], leading to widespread impacts across the Pacific. In negative phases of the longer-period PDO, the south-western Pacific is warmer than in positive phases, leading to higher rainfalls.

### 3.7. Changing Climate and Rainfall Trends

Projections for future climate in the 21st century based on the Coupled Model Intercomparison Project Phase 5 using global climate models have been made for Tonga under various Representative Concentration Pathways estimating possible future trends in greenhouse gas emissions [21]. These projections suggest that there will be more extreme rainfall events, continuing sea level rise, increasing ocean acidification and higher risk of coral bleaching, and that El Niño and La Niña events will continue (all very high confidence). It is projected that the frequency of tropical cyclones will decrease by the end of the 21st century (medium confidence) but their maximum intensity may increase [34,35]. There is no consensus on whether average annual rainfall will increase or decrease (low confidence), and it is projected that drought frequency may slightly decrease (low confidence) [21].

The CSIRO and Australian Bureau of Meteorology [21] have provided no projections of trends in potential evaporation or actual evapotranspiration. Earlier projections based on increasing temperatures [33] are erroneous. The available climate records for Tonga show average atmospheric temperatures increasing by about 1 °C per century [14], similar to increases in sea surface temperatures in the surrounding ocean. Table 5 lists the historic linear regression trends in annual and seasonal rainfall at the meteorological stations in Table 1.

There are no significant trends in Table 5 in annual or wet season rainfall over the period from 1947 to 2019. This is consistent with climate change projections [21]. Out of the four stations with long-term rainfall records, only Lupepau'u in Vava'u showed a significant increasing trend in dry season rainfall over this period. In contrast, the two stations with shorter rainfall records, i.e., 1980 to 2019, in the northern and southern island Divisions had significant and identical—within the margin of error—increasing trends in both annual rainfall and wet season rainfall. While it is tempting to attribute these more recent increasing trends over 39-year periods to climate change, they are very strongly negatively correlated with trends over 31 years in the PDO, unrelated to changing climate (Figure 4).

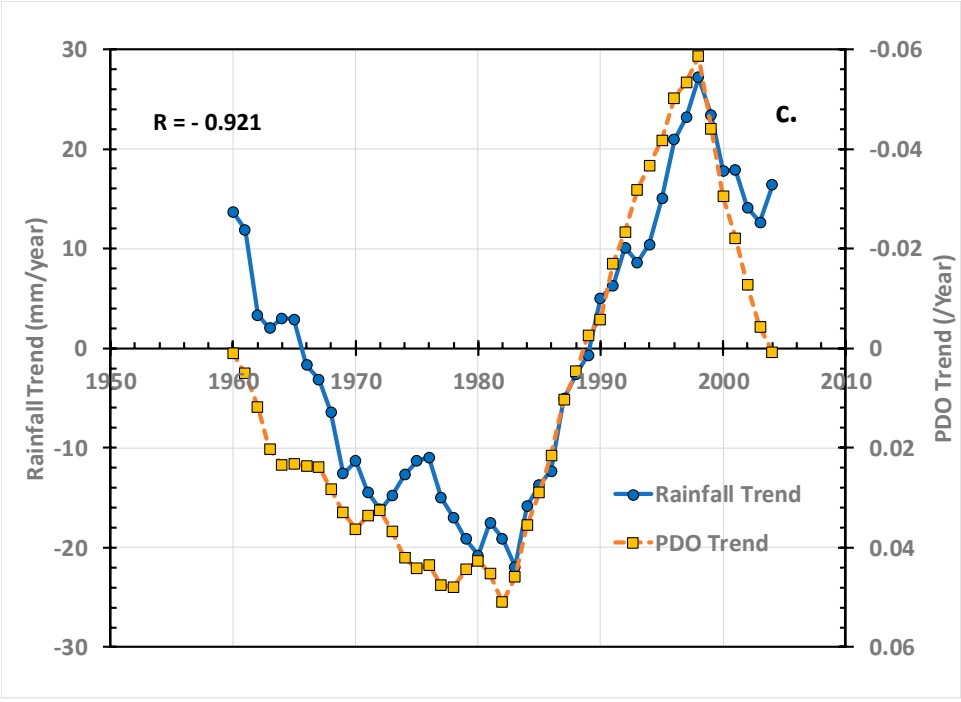

**Figure 4.** Comparison between the trends in rainfall at Nuku'alofa over 31 year consecutive periods and the consecutive trends in the PDO over the same period. The correlation, −0.921, is highly significant. The year shown is the midyear of the 31-year period.

**Table 5.** Historic linear regression trends in annual and seasonal rainfalls with standard errors and significance level (Signif) for the meteorological stations in Table 4. Long- and short-term records are compared over the same periods, i.e., 1947–2019 and 1980 to 2019, respectively. Bold values are statistically significant.

| Met Station | Trend Annual Rainfall (mm/Decade) | Signif | Trend Nov-Apr Rainfall (mm/Decade) | Signif | Trend May-Oct Rainfall (mm/Decade) | Signif | Period |
|---|---|---|---|---|---|---|---|
| Niuafo'ou | **204 ± 71** [1] | ***p < 0.05*** | **82 ± 36** | ***p < 0.05*** | **141 ± 68** [1] | ***p < 0.05*** | 1980–2019 |
| Niuatoputapu | −3 ± 30 | NS | −9 ± 23 | NS | 6 ± 19 | NS | 1947–2019 |
| Lupepau'u | 34 ± 28 | NS | 2 ± 24 | NS | **34 ± 14** | ***p < 0.05*** | 1947–2019 |
| Ha'apai | 14 ± 25 | NS | 3 ± 22 | NS | 14 ± 12 | NS | 1947–2019 |
| Fua'amotu | **212 ± 57** | ***p < 0.001*** | **162 ± 58** | ***p < 0.01*** | 60 ± 35 | NS | 1980–2019 |
| Nuku'alofa | 0 ± 24 | NS | −7 ± 21 | NS | 9 ± 12 | NS | 1947–2019 |

[1] Period 1981–2019, data missing for 1980.

### 3.8. Droughts

Because of the nation-wide reliance on rainwater harvesting or relatively shallow groundwater, droughts are particularly consequential in Tonga. They are projected to slightly decrease in frequency under climate change (low confidence) [21]. Projections, however [37–40], suggest that the frequency of extreme ENSO events will increase from one in 20 years to one in 10 years within the 21st century, and the PDO will become less predictable [41]. This implies that severe droughts in Tonga may increase in frequency. Resident household sizes and their water demands mean that rain tanks supplying households often fail during protracted dry seasons or droughts. Table 6 lists the historic frequency of severe, i.e., 6-month, dry periods in which the average rainfall per month was less than or equal to 60 mm and when household rain tanks are predicted to run dry.

**Table 6.** Frequency of severe dry periods in both wet and dry season at meteorological stations in Tonga.

| Met Station | Frequency Severe Dry Periods | | Period |
|---|---|---|---|
| | Nov.–Apr. | May–Oct. | |
| Niuafo'ou | - | 1/46 | 1980–2019 |
| Niuatoputapu | - | 1/43 | 1947–2019 |
| Lupepau'u | - | 1/28 | 1947–2019 |
| Ha'apai | - | 1/9 | 1947–2019 |
| Fua'amotu | 1/40 | 1/26 | 1980–2019 |
| Nuku'alofa | 1/39 | 1/28 | 1947–2019 |

In terms of dry season rainwater shortages in the island Divisions, the frequency in Table 6 increases in the order of Ongo Niua < Vava'u = Tongatapu < Ha'apai. The frequency of expected raintank failures in Ha'apai is three times that in Tongatapu/Vava'u and five times that in Ongo Niua. The frequency of expected rain tank failures in the periods from 1980 to 2019 and 1947 to 2019 at Fua'amotu and Nuku'alofa on Tongatapu are similar. This is also the case for the lower failure frequencies over these two periods at Niuafo'ou and Niuatoputapu in Ongo Niua. These suggest that, at present, there is no discernable impact of climate change on drought frequency. Surprisingly, Table 6 also indicates that, unlike other Island Divisions, the wet season in Tongatapu will also fail to supply adequate rainfall for household supply about once in forty years. Failure of the wet season is a serious concern. These results suggest that both Hapai and Tongatapu have higher risks of rainwater harvesting supply failures than the other island Divisions.

The larger water storages in groundwater systems mean that they are more robust supply sources during droughts, provided groundwater extraction is carefully monitored and managed [14].

Census and hydrological data have revealed major differences in access to acceptable and reliable sources of water across Tonga, with no overall long-term trends in rainfall, despite shorter-term trends driven by the PDO variations.

*3.9. The Tonga Strategic Development Framework 2015–2025*

The seven planned National Outcomes (NOs) of TSDFII are listed in Table 7. The five institutional and input pillars, together with their accompanying OO and the number of associated SCs, are listed in Appendix A. Infrastructure services are incorporated within NO E in Table 7. In total, 14% of the planned NOs concern infrastructure. The OO under the Infrastructure and Technology Inputs Pillar, associated with NO E, focus on more reliable, safe and affordable energy, transport and information and communications technology (ICT), building and structures and research and development services, but they do not include water supply services. Table 8 compares the number of OOs and SCs devoted to infrastructure services and compares them with those targeting water supply.

**Table 7.** The seven planned National Outcomes of TSDFII [12].

| Code | National Outcome |
|:---:|:---|
| A. | A more inclusive, sustainable and dynamic knowledge-based economy |
| B. | A more inclusive, sustainable and balanced urban and rural development across island groups |
| C. | A more inclusive, sustainable and empowering human development with gender equality |
| D. | A more inclusive, sustainable and responsive good governance with law and order |
| E. | A more inclusive, sustainable and successful provision and maintenance of infrastructure and technology |
| F. | A more inclusive, sustainable and effective land administration, environment management, and resilience to climate and risk |
| G. | A more inclusive, sustainable and consistent advancement of our external interests, security and sovereignty |

**Table 8.** The number of Organizational Outcomes and Strategic Concepts assigned to services in TSDFII. The total number of each are in parentheses.

| Service | Organizational Outcome (29) | Strategic Concepts (153) |
|:---:|:---:|:---:|
| Energy | 1 | 4 |
| Transport | 1 | 9 |
| Information & Communications | 1 | 9 |
| Building & Structures | 1 | 5 |
| Research & Development | 1 | 8 |
| Water Supply | 0 | 1 |

Table 8 shows that over 17% of all OO in TSDFII are assigned to infrastructure services, an important sector of the government's agenda, but water supply is not included in any OO. Nearly 24% of all SCs highlight infrastructure services, with transport and ICT each having nearly 6% of SCs. Water has nearly an order of magnitude less, i.e., 0.7%, with only one mention under the environment and natural resources pillar: "improve the management and delivery of safe water supply for business and households". Under the Health component of the Social Institutions Pillar, "Percentage of population with safe water supply" is listed as a key performance indicator (KPI), but there is no OO or SC associated with this KPI), nor any associated with environment and natural resources. This means that government performance in managing and delivering water supply at the Island Division level cannot be measured.

In mapping the planned National Outcomes against the UN SDGs, the TSDFII identifies National Outcomes F, E, and B (Table 7) as contributing to the UN's SDG6, yet the Infrastructure and Technology Inputs Pillar associated with National Outcome E does not mention water supply services or sanitation. In summary, not one of the 29 OOs identifies water supply as a nationally important infrastructure service. Only one of the SC raised in consultations mentions safe water supply, and nowhere is the adequacy of water supply raised, despite the large differences between island Divisions found in Table 2 and shown in Figure 2.

*3.10. Village Community Development Plans, 2016*

Table 9 provides details of the number of village CDPs that were available for analysis for each of the Island Divisions and Districts in Tonga. In total, 117 CDPs were available of the original 136 that were presented in 2016. CDPs for the main island Tongatapu excluded the Districts of Kolofo'ou and Kolomotu'a, that comprise the capital area, Greater Nuku'alofa, and so represent rural areas of Tongatapu. Each village identified a different number of priority issues to be addressed. The number varied among Island Divisions and among women, youths and men.

**Table 9.** The number of accessed village community development plans for Districts and Island Divisions in Tonga and the medium number of identified priorities identified by women, youths and men within each Island Division [27].

| Island Division | District | No. of Village CDPs | No. CDPs/Island Division | Median Number of Priorities in Island Division | | |
|---|---|---|---|---|---|---|
| | | | | Women | Youth | Men |
| 'Eua | 'Eua Motu'a | 13 | 13 | 8 | 7.5 | 7.5 |
| Tongatapu | Nukunuku | 9 | 48 | 6 | 5 | 6 |
| | Tatakamatonga | 7 | | | | |
| | Vaini | 11 | | | | |
| | Lapaha | 10 | | | | |
| | Kolovai | 11 | | | | |
| Vava'u | Hahake | 8 | 39 | 7 | 6 | 7 |
| | Hihifo | 5 | | | | |
| | Leimatu'a | 4 | | | | |
| | Pangaimotu | 4 | | | | |
| | Motu | 9 | | | | |
| | Neiafu | 9 | | | | |
| Ha'apai | | 5 | 5 | 9 | 7 | 10 |
| Ongo Niua | Niuatoputapu | 4 | 12 | 5 | 5 | 5.5 |
| | Niuafo'ou [1] | 8 | | | | |
| Total | | 117 | Country Median | 6 | 6 | 6 |

[1] One village in Niauafo'ou gave aggregated priorities rather than separate women's, youths' and men's priorities.

For the whole country, Table 9 shows that the median numbers of development priority issues identified in villages by women, youths and men were the same, i.e., six each. Ongo Niua Island Division identified the fewest priority issues, while Ha'apai Island Division identified the most. This may be a result of the impact of TC Ian on Ha'apai in 2014. Youths in 'Eua identified slightly more priority issues than youths elsewhere. The maximum numbers of priorities identified by women, youths and men were 12, 11 and 12, respectively, while the minimum numbers were 4, 3 and 3.

Table 10 lists the percentage of villages in each Island Division that identified water supply issues as their number one priority. Over 56% of the available village CDPs throughout Tonga and over 53% in all individual Island Divisions identified water supply as the number one priority. Women in Tonga ranked water supply as a higher priority issue than did men or youths. Only in 'Eua did youths and men view water supply as a higher priority than did women. On 'Eua, youths had the highest concern about water supply among the youths in any Island Division. Women in the Ha'apai Division had the greatest concerns (80%) about water supply of any gender-age cohort and Island Division.

**Table 10.** The percentage of villages in each Island Division that identified water supply issues as first priority in terms of women's, youths' and men's perspectives.

| Island Division | Percentage of Villages with Water Supply as Highest Priority | | | |
|---|---|---|---|---|
| | Women | Youth | Men | Average |
| 'Eua | 46% | 62% | 54% | 54% |
| Tongatapu | 67% | 42% | 54% | 54% |
| Vava'u | 67% | 49% | 62% | 59% |
| Ha'apai | 80% | 60% | 20% | 53% |
| Ongo Niua | 50% | 58% | 50% | 53% |
| **Tonga** | **63%** | **49%** | **55%** | **56%** |

Figure 5a shows the distribution of the percentage of villages throughout the Island Divisions where concerns over water supply were ranked within the top three priorities. The percentage of villages in Tonga that ranked water supply in the top three priorities was 76%, with women comprising a slightly higher percentage (83%) than men (80%), and considerably higher than youths (66%). Concerns over village water supply were highest in Ha'apai (93%), where both 100% of villages in the women and youths cohort ranked water within the top three priorities compared with 80% for men. Villages in the wetter, northern Ongo Niua Division had the lowest average percentage, but still 72% ranked water supply issues within the top three village priorities. Youth in Ongo Niua villages had the lowest percentage (58%) of any group who ranked water within the top three priorities.

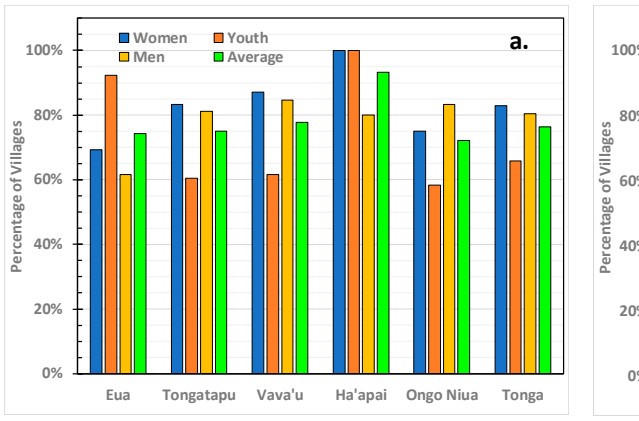 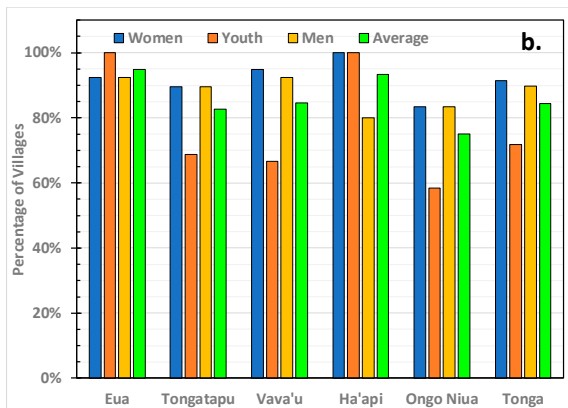

**Figure 5.** The percentage of villages throughout Tonga's Island Divisions, as well as Tonga collectively, which ranked water supply: (**a**) within the top three priorities; and (**b**) as a priority in terms of women's, youths' and men's perspective and the village average of all three.

Figure 5b shows the number of villages that ranked water supply as one of their development priorities. Overall, 84% of villages in Tonga listed water supply as a priority. The least concern among Island Groups over water supply was observed in the main island of Tongatapu (83%). Youth in Ongo Niua were the group with the lowest level of concern (58%) among all groups about water supply. The greatest concerns were in Ha'apai, where 100% of women and youths ranked water supply as a priority, as did youths in 'Eua. Nationally, women in villages were slightly more concerned, with 91% listing water as a priority, in contrast to 90% of men; both sexes had more concerns than youths (72%).

The correlations between the percentage of villages in Island Divisions that ranked water supply as their top priority, within their top three priorities or as a priority and the WSR of Equation (1) were not significant ($p < 0.1$). This means that, although the census data show wide variation in the water sources available to different Divisions, they are not reflected in differences in their priority concerns over water supply.

## 4. Discussion

### 4.1. Census, Hydrological Data and Water Supply

The census data in Table 1 show the concentration of population of Tonga in both the urban and rural areas of the main Tongatapu Island Division, which includes the capital, Nuku'alofa. Just over a quarter of the 26% the population is dispersed across villages in the other four Island Divisions. The overall population has declined although inward migration to the main island and the capital is occurring, some of which may have been the result of TC Ian which devastated Ha'apai in 2014. This inward migration has been a continuing feature of multi-island PICs for several decades [42].

Table 2 reveals that Tongans switch water sources between public piped groundwater supply and private household rainwater harvesting depending on water use. Rainwater is preferred for drinking. The greater volumetric use of water for nondrinking purposes, such as toilet flushing washing, and bathing means that the piped water supplies are important. We used here the ratio of the number of households accessing rainwater supply to those accessing piped water supply, Equation (1) as a coarse measure of the reliability of supply. The results show urban areas and population centers have greater access to piped groundwater supply than in rural areas with very limited use of private local groundwater wells which are prone to contamination from household sanitation systems.

The WSR of Island Divisions (Figure 2) identified that Ha'apai, Ongo Niua and Vava'u Island Divisions are vulnerable because of their larger reliance on drought-sensitive household rainwater harvesting water supply systems. Outer islands in these three Island Divisions are more reliant on rainwater harvesting, and from the few examples in Table 3 have extremely limited average daily per capita water use about equal to the World Health Organization's recommended minimum quantity of water required to satisfy essential health and hygiene needs in emergency situations [17]. Estimations of the frequency of failure of rain tank systems based on available rainfall data (Table 4) reveal that even well-managed rainwater harvesting systems will fail periodically especially in the May to October drier season. Ha'apai is more vulnerable to rainwater system failure than southern or northern islands (Table 5). The diversity in access to water sources of different reliability throughout Tonga's islands, as evidenced by WSR (Equation (1)), means that Target 6.1 of UN SDG6 of achieving universal and equitable access to safe and affordable water for all remains a major challenge.

The greatest volumetric use of water in Tonga is sourced from groundwater. In the southern Island Divisions of 'Eua and Tongatapu, water is sourced from springs and well-studied relatively thick freshwater lenses with comparatively low salinity and significant water yields. In the northern Island Divisions of Vava'u and Ha'apai, fresh groundwater lenses are much thinner, and the salinities of water supplied to population centers are more saline than in Tongatapu and can be even brackish in droughts with limited groundwater yields. In other islands, the availability of groundwater for water supply remains to be determined.

This work has sought to address three questions. The first was: In the absence of detailed water data, can census data, together with available hydrological data, be used to asses planning priorities for water supply in dispersed island countries? We have sought to answer this question at the Island Division level. It was found that census data and the limited hydrological data available identified significant differences in water availability that can be used to better inform planning processes.

Rainfall is centrally important to groundwater recharge, piped water and rainwater supply systems in Tonga. The distribution and variability of rainfall in Tonga (Table 4) differs across Tonga's five Island Division. Both depend on proximity to the SPCZ (20, 33) and on major ocean-atmosphere interactions, ENSO events and the PDO. Projections on the impact of climate change on annual rainfall and drought frequency and intensity are equivocal and of low confidence [21,33]. While air and ocean temperatures have increased by about 1 °C over the period from 1947 to 2019, the historic record shows no statistically significant long-term trends in annual rainfall (Table 5). There are shorter-term trends in rainfall (Table 5) over 31-year periods which are highly significantly negatively correlated with trends in the PDO over the same period (Figure 4).

The second question this work has sought to address is: Do variations in rainfall due to climate change need to be addressed for water supply in medium-term (10 years) national development plans? We have shown here in Tonga that the historic increase in temperatures over the period from 1947 to 2019 has not be associated with any historic increase in annual rainfall or its variability. Since both are key factors in water supply in Tonga, the answer here, compared with the variability imposed by ENSO and PDO, is that climate change is of secondary importance in decade-long development plans addressing water supply. The conclusion is consistent with findings that other factors, including governance, increasing demand and urbanization pose much greater shorter-term risks to water supply in PICs [43]. Continued impacts of tropical cyclones on water supply infrastructure, especially supply and rain tanks, can be devastating. The change in TC frequency and intensity with climate change [21,33–35] could pose increased risks to water supply systems as would climate change induced alterations to the frequency and intensity of ENSO and PDO events [36–41].

### 4.2. Top-Down versus Bottom-Up Planning Priorities

The largely top-down TSDFII [12] involved discussions with key sectors of the economy including District and Town officers, church leader forums, nongovernment organizations and all main sectors of private business forums over a three-month period. TDSFII claims that the planned National Outcomes, F, E and B (Table 7) are contributions to the UN's SDG6 on water and sanitation. Water and sanitation do not appear in any of the Organizational Outcomes associated with National Outcomes F, E or B.

Investment in infrastructure is a major driver of economic productivity and development [44]. Our analysis of the weight given to infrastructure services in Table 8 shows that while services are assigned to nearly 17% of the total 29 OO, water supply was not mentioned. Of the 153 Strategic Concepts raised in consultations leading to TSDFII, infrastructure services were raised in nearly 24% of SCs, with transport and ICT services each identified in nearly 6% of SCs. In contrast, water supply services ranked nearly an order of magnitude lower in SCs, i.e., at 0.7%. TSDFII conveys the strong impression that water supply is a low national priority in Tonga compared to other services.

The rural village, bottom-up, Community Development Plans in Tonga evolved over a 9-year period, partly because each stage of the planning process required participation of 80% of the population of each village, and partly because of the dispersal of villages over Tonga's many villages. The available 117 CDPs, analyzed for the first time here, represents 77.5% of the total number of rural villages in Tonga and 86% of the 136 CDPs presented in 2016. The greatest number of available CDPs come from the main island Tongatapu, followed by the Vava'u Division, reflecting the distribution of the rural population (Table 1). The least number were from Ha'apai, which may reflect the devastation caused by TC Ian in 2014.

The median number of priority issues ranked in the CDPs for Tonga was six, the same across gender and age groups. The highest number of priorities identified was in the Ha'apai Division, and the lowest number was in the wetter, northern Ongo Niua Division (Table 9). Over 56% of all rural villages ranked water supply as their top priority, with women ranking it as the first priority more frequently than youths or men (Table 10). This reflects the fact that in rural villages, household water supply is largely the responsibility of women. In Ha'apai, women in 80% of the villages ranked water supply as the first priority, in contrast to men in Ha'apai villages, where only 20% ranked it as the highest priority.

Around 76% of all villages ranked water supply within the top three priority issues, with women giving a higher ranking than men, followed by youths (Figure 5a). The highest rankings were in villages in Ha'apai, with women and youths in 100% of villages ranking water supply within the top three priorities. The lowest top three ranking was in the Ongo Niua Division, due to low rankings by youths. Even in rural Tongatapu, 75% of villages ranked water supply within the top three priorities. For Tonga as a whole, 84% of villages included water supply within their village priorities with the lowest Island Division being Ongo Niua, 75%, and the highest being 'Eua, 95%. In Ha'apai, 100% of women and youths as well a 100% of youths in 'Eua, in all villages ranked water supply as a development priority

(Figure 5b). These overwhelming local village development priorities for improved water supply are in sharp contrast to very low concern over water supply in TSDFII.

One hypothesis was that village priorities in water supply may be linked to the WSR from census data as a coarse measure of the accessibility of water for nondrinking purposes. There was no statistically significant ($p > 0.1$) correlation between village water supply priorities and WSR from the census data. There is an inbuilt assumption in WSR that all Island Division piped water supplies are equivalent in adequacy and quality. While the capital area, Greater Nuku'alofa, has access to continuous piped water supply, village piped water supplies are almost all intermittent, even on the main island, Tongatapu. Moreover, the groundwater supply in Tongatapu is relatively fresh, while those in Vava'u and Ha'apai have higher salinity and can even be brackish during droughts. The wide-spread concerns over water supply expressed in the village CDPs involve supply from rain tanks, and village piped water supply.

The third question we have attempted to answer is: Do top-down governance templates, such as national sustainable development strategies, capture the priorities of rural island communities in water supply? From our analysis of TSDFII and CDPs, the answer in Tonga is an emphatic "NO!".

### 4.3. Possible Reasons for the Mismatch in National and Local Planning Priorities

There is an enormous difference in the priorities given to water supply in the top down TSDSII compared with the bottom up CDPs. Traditionally in many PICs, water supply was a responsibility of the family or extended family. In Tonga, water supply is still a shared responsibility between households with their private rainwater harvesting systems and authorities who supply piped water. It may be that the ministries that developed TSDFII do not see water as a government responsibility, compared to other infrastructure services. The recent passage by Parliament of the Tonga Water Resources ACT 2020 may change that view.

Another factor could be that the ministries, departments and agencies involved in developing TSDFII are mostly based in Greater Nuku'alofa, which has a continuous, very adequate per capita piped supply (Table 3) of lower salinity groundwater, so water supply for them is not a priority.

A final factor could be that prior to 2020, there was no ministry with overall legal responsibility for managing, conserving and protecting the nation's water resources. In effect, water resources did not have a voice in the national planning process.

### 4.4. Limitations of This Work

Comparison between national and local development plans involved assumptions about priorities given to different sectors. We have only analyzed water supply and not all sector priorities in the CDPs. In an attempt to remove bias in our analysis of TSDFII, we analyzed the national priority given to water supply services relative to other infrastructure services (Table 8). The omission of water supply services is thus put into the context of other infrastructure services which are necessary for development.

This work only analyzed census and CDP data at the Island Division level to facilitate a comparison with the national level TSDFII. Both contain a wealth of valuable information at the District and village levels which could better inform planning processes. We compared the lack of emphasis on water supply in TSDFII with water supply data in the 2016 census and water priorities in village CDPs. TSDFII was produced in 2015, while CDPs were presented in 2016, and the 2016 census results were not available until late 2017. Examination of the 2006 census results, which were available in 2008 in plenty of time to be incorporated in TSDFII, show even larger challenges in water supply than in the 2016 census. Work on the CDPs commenced in 2007, and widespread concerns about water supply have been evident for decades [24].

WSR, Equation (1), was used as a coarse measure of the ease of accessing appropriate quantities of reliable water supplies throughout Tonga's Island Divisions. This assumes that piped water supplies are more reliable, are of larger volume than rainwater harvesting supplies and that piped supplies are similar in all Island Divisions. The lack of correlation between WSR and Island Division level

water supply priorities indicates that intermittent piped water supplies, a feature of village water supply systems, salinity of water supply and the cost of local water supply may influence village priority rankings.

## 5. Conclusions

Better governance has been proposed as a key strategy for improving water security in PICs. National Sustainable Development Strategies have been promulgated as an improved overarching governance instrument to identify national priorities, plan their solutions, identify responsibilities, allocate resources, monitor and evaluate outcomes, as well as fulfil international and regional commitments, especially the UN SDGs. The relevance of transplanting governance instruments in the Pacific has been questioned, as has the applicability of time-efficient, top-down planning processes to local village priorities, particularly in terms of water supply. Many SIDs, with dispersed island communities, have a wide range of local conditions that need to be accommodated by planning instruments, as well as the differing concerns of women, youths and men. The Kingdom of Tonga presents the unique opportunity to contrast priorities given to water supply in the top-down Tonga Strategic Development Framework 2015–2025, developed after three months of high-level consultations, with water supply priorities identified in 9 years of community consultations that led to Community Development Plans for rural villages through Tonga's five Island Divisions. Priorities in these CDPs appear not to have been analyzed previously. They contain a wealth of information which could better inform priorities in revising TSDFII.

The TSDFII analysis revealed that, although improved infrastructure services were a significant proportion of planned organizational outcomes, water supply was not included. This implies that water supply is not a national priority, and that TSDFII, therefore, does not address SDG6. In contrast, 84% of villages ranked water supply as a priority, and 56% of villages ranked water supply as their highest priority. Island Divisions with highest concerns were clearly identified, as were the contrasting priorities of women, youths and men. Since NSDS have been widely promulgated throughout PICs, the mismatch found here in national and local water supply priorities warrants investigation in other island countries. It also points to the importance in multi-island countries of investing in bottom-up planning processes, building on the strengths of island communities.

In many PICs, information on water sources and their uses is limited. In this work, census results and the available hydrological data were used to contrast differences in Island Division water supplies. There were large differences between the capital area, main island and outer islands. Census and hydrological data, no matter how limited, are valuable for better informing priorities in NSDS processes. The analysis also showed that in Tonga, over the 10-year planning horizon, climate change was not a significant factor compared with the variability imposed by ENSO and PDO events. The impact of climate change on the frequency and intensity of extreme events, such as droughts and tropical cyclones, which impact water supply, remains a pressing research question.

The lack of correlation between the census-derived water source ratio for nondrinking purposes and Island Division CDP water supply priorities identified in all Island Division CDPs is interesting, since all water sources have been deemed safe for drinking. The overwhelming concern in village communities appears to be the adequacy and reliability of water supply for higher volume usage, i.e., bathing, washing, sanitation and hygiene. Village per capita water supplies from rainwater harvesting systems of around 20 L/pers/day, found here in the limited number of outer islands with data available, are not adequate for these purposes.

A final question raised in this work was: Can national development plans be improved for water supply in multi-island countries? Improved governance and increased water information may be keys to better water management, but governance instruments such as overarching NSDS need to be adapted for use in PICs. They need to be informed by available census and hydrological data and by the results of continuing local planning processes, for which island communities have considerable strengths and

long experience. Local processes may be time-consuming, but as shown here, more efficient top-down processes can fail completely to capture widespread local development priorities.

The Fale Alea 'o Tonga (National Parliament) has very recently passed the Water Resources Act 2020 to conserve, protect and manage the Kingdom's water resources. This requires, amongst other objectives, the "implementation of urban and rural planning regimes that take account of water management". This has the potential to address gaps in knowledge of the Kingdom's water resources and their use, and to focus on improving water supplies in Island Divisions and villages identified in the CDPs. It also provides a stimulus to revise TDSFII to better reflect island community concerns.

**Supplementary Materials:** The following are available online at http://www.mdpi.com/2306-5338/7/4/81/s1, Table S1: Summary Water Priorities in Community Development Plans. Rural Villages in Tonga.

**Author Contributions:** Conceptualization, I.W., T.F., and T.K.; methodology, I.W., T.F., and T.K.; formal analysis, T.F. and I.W.; writing—original draft preparation, I.W.; writing—review and editing, I.W., T.F., and T.K.; project administration, T.K. All authors have read and agreed to the published version of the manuscript.

**Funding:** This research received no external funding.

**Acknowledgments:** This work arose as a result of a strategic planning project supported by the World Meteorology Organization under its Climate Risk and Early Warning Systems Initiative for Pacific Island Countries focusing on improved governance. WMO Pacific are thanked for that support. We are grateful to Quddus Fidelia, Tonga Water Board, for providing helpful information on urban water supplies in Tonga, to Lopeti Tufui, Victoria University of Wellington, for information on rural groundwater pumping in Tongatapu and Karin Nagorcka for editorial assistance. We thank an unnamed reviewer for helpful comments which improved this paper.

**Conflicts of Interest:** The authors declare no conflict of interest.

## Appendix A

**Table A1.** The Pillars and associated Organisational Outcomes and number of Strategic Concepts connected with each OO in the TSDFII [12].

| Pillars | Organisational Outcomes | No. Strategic Concepts |
|---|---|---|
| 1. Economic Institutions | 1.1 Improved macroeconomic management & stability with deeper financial markets | 6 |
| | 1.2 Closer private/public partnerships for economic growth | 4 |
| | 1.3 Strengthened business enabling environment | 4 |
| | 1.4 Improved public enterprise performance | 5 |
| | 1.5 Better access to, & improved use of overseas trade and employment, & foreign investment | 5 |
| 2. Social Institutions | 2.1 Improved collaboration with and support to civil society organizations & community Divisions | 7 |
| | 2.2 Closer partnerships between government & churches & other stakeholders for community development | 4 |
| | 2.3 More appropriate social & cultural practices | 7 |

| | | |
|---|---|---|
| | 2.4 Improved education & training providing lifetime learning | 11 |
| | 2.5 Improved health care & delivery systems (universal healthcare coverage) | 5 |
| | 2.6 Stronger integrated approaches to address both to address both communicable and noncommunicable diseases | 4 |
| | 2.7 Better care & support for vulnerable people, in particular the disabled | 6 |
| | 2.8 Improved collaboration with the Tongan diaspora | 3 |
| 3. Political Institutions | 3.1 More efficient, effective, affordable, honest, transparent & apolitical public service focussed on clear priorities | 7 |
| | 3.2 Improved Law & order & domestic security appropriately applied | 7 |
| | 3.3 Appropriate decentralisation of government admin. With better scope for engagement with the public | 3 |
| | 3.4 Modern & appropriate Constitution with laws & regulations reflecting international standards of democratic processes | 4 |
| | 3.5 Improved working relations and collaboration between Privy Council, executive, legislative & judiciary | 2 |
| | 3.6 Improved collaboration with development partners ensuring programs better aligned behind govt. priorities | 4 |
| | 3.7 Improved political & defence engagement within the Pacific & the rest of the World | 4 |
| 4. Infrastructure & Technology Inputs | 4.1 More reliable, safe & affordable energy services | 4 |
| | 4.2 More reliable, safe & affordable transport services | 7 |
| | 4.3 More reliable, safe & affordable information & communication technology (ICT) used in more innovative ways | 4 |
| | 4.4 More reliable, safe & affordable buildings & other structures | 5 |
| | 4.5 Improved use of research & development focussing on priority needs based on stronger foresight | 3 |
| 5. Natural Resources & Environment Inputs | 5.1 Improved land use planning, administration & management for private and public use | 7 |
| | 5.2 Improved use of natural resources for long-term flow of benefits | 7 |
| | 5.3 Cleaner environment with improved waste recycling | 5 |
| | 5.4 Improved resilience to extreme natural events and climate change | 9 |

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
