# Peer review of "Meeting SDG6 in the Kingdom of Tonga: The Mismatch between National and Local Sustainable Development Planning for Water Supply"

_hydrology, doi:10.3390/hydrology7040081_

Round 1

Reviewer 1 Report

GENERAL COMMENTS

This manuscript presents a study on the priorities in water supply as they are defined by the bottom-up community development plans and the top-down Tonga Strategic Development Framework, and discusses on the mismatch between these two processes. The manuscript is interesting, well written and structured. Some minor issues are highlighted in the specific comments.

SPECIFIC COMMENTS

Location: Line 60, " ...supply identified by both top-down and bottom up planning processes ..."
Comment: The "bottom up" should be "bottom-up".

Location: Line 71, Line 133
Comment: What is "CPDs"?

Location: Line 107-109
Comment: These lines are not clear. An equation would help clarify the concept.

Location: Table 1
Comment: This table is not well-structured. The column "Tonga" appears to be the total population. If so, it should be appear as last column with a name like "Tonga Total" or "Tonga Overall". "Urban" and "Rural" columns should become "Urban areas", "Rural areas". The columns "Eua", "Tongatapu", "Vava'u", etc., should become "Division of Eua", etc.

Location: Table 2
Comment: See previous comment.

Location: Figure 2
Comment: See previous comment.

Location: Lines 435-443, Figure 5
Comment: No additional information is provided by providing statistics for the top three priorities. This paragraph and Figure 5 may be removed.

Author Response

Responses to Review No 1

Thanks for these they have helped improve the manuscript

SPECIFIC COMMENTS

Location: Line 60, " ...supply identified by both top-down and bottom up planning processes ..."

Comment: The "bottom up" should be "bottom-up".

Changed

Location: Line 71, Line 133

Comment: What is "CPDs"?

Should be CDPs

Location: Line 107-109

Comment: These lines are not clear. An equation would help clarify the concept.

Equation (3) now included

Location: Table 1

Comment: This table is not well-structured. The column "Tonga" appears to be the total population. If so, it should be appear as last column with a name like "Tonga Total" or "Tonga Overall". "Urban" and "Rural" columns should become "Urban areas", "Rural areas". The columns "Eua", "Tongatapu", "Vava'u", etc., should become "Division of Eua", etc.

The Table is presented as in the Census results. The suggested changes would clutter the Table. We have made the suggested changes in the Table caption.   

Location: Table 2

Comment: See previous comment.

Suggested changes made in the Table caption

Location: Figure 2

Comment: See previous comment.

Suggested changes made in the Figure caption

Location: Lines 435-443, Figure 5

Comment: No additional information is provided by providing statistics for the top three priorities. This paragraph and Figure 5 may be removed.

We disagree with this comment. We are attempting to show the level of priority giving to water supply in villages. We show the number that give it top priority, the number that give it within their top three priorities and then the number that list it as a priority. This demonstrates the importance given to water supply by villagers.

Reviewer 2 Report

The manuscript presents the analyses of the GAP between national and local sustainable development planning for water supply in the Kingdom of Tonga.

Mentioning SDG6 in the title should be omitted because only a minor part of the manuscript refers to it (few sentences in Tittle, Introduction, 3.8. in Results and  4.2. in Discussion).

Although the collection of data and the analyses are presented well, and are very interesting, there is no additional new knowledge or methodology developed. How can the facts derived from analyses and presented in conclusion improve the water supply in Tonga or in similar countries?

Scientific papers should present new knowledge, methods, methodologies that should help improving (in this case) the water management and sustainable development planning. The paper written in this way has no scientific added value to water management.

In the Introduction it is important to analyze the existing knowledge, methods and methodologies regarding the topic of the article:

  • Do similar problems of differences between national and local planning regarding water supply exists in other SIDSs? Are there any articles that cover this problem or the problem of prioritizing the water supply?
  • How did you define the methodology that was used in this manuscript? Are there other methodologies that can be used for analyzing the differences between national and local planning (literature overview).

I suggest to rewrite the paper in the way to stress on:

  • The methodology that has been used for the analyses (present on flowchart, explain in text)
  • How it can help in defining the GAP on different levels of planning especially for SIDS?
  • What are the recommendations for improving the water management and sustainable development planning (lessons learned)?
  • What are the suggestions to improve the water supply in the Kingdom of Tonga, based on identified GAP?

Author Response

Response to Reviewer No 2

The authors are grateful for the reviewer’s comments. They have helped to improve significantly the manuscript.

  1. Improve the introduction (state of the art/research)

The introduction has been improved and includes a summary of the state of the art/research:

  1. Pacific island countries (PICs) in the Oceania region failed to meet the 2015 Millennium Development Targets
  2. Poor governance has been advanced as one of the reasons for this failure another has been impacts of climate change, and a third is inadequate information on water resources
  3. Top-down National sustainable development plans have been introduced into PICs by the UN and major development banks as the ultimate overarching national governance instrument to address all national priorities and to fulfill commitments to international and regional agreements such as the 2030 UNSDGs which includes SDG6.
  4. The relevance of imported governance instruments to PICs has been questioned, as has the relevance of top-down processes.
  5. Tonga provides a unique opportunity to compare priorities assigned by this top-down process with the results of a national village-level bottom-up consultation process which ranked priorities for women, youth and men.

The paper therefore seeks to answer 3 questions (now specified in the introduction):

  • In the absence of detailed water data, can Census data, together with available hydrological data, be used to asses planning priorities for water supply in dispersed island countries?
  • Do variations in rainfall due to climate change need to be addressed for water supply in medium term (10 years) national development plans?
  • Do top-down governance templates, such as national sustainable development strategies, capture the priorities of rural island communities in water supply?
  1. Define the methodology for analysing the GAP (mismatch) between national and local planning (stressing on water supply).

This is an excellent point. The Materials and Methods section has now been improved to explain the gap analysis. In TSDFII the key national priorities are identified in 29 national priority planned Organisational Outcomes. We now compare the weight given to water supply in OOs compared to other infrastructure services.  In addition, we rank the weight given to water supply in the Strategic Concepts which were points raised during consultations. Although the combined infrastructure services were 17% of the total 00, water supply was 0%. Water ranked 0.7% of the 153 SCs almost an order of magnitude smaller than other services (now included as Table 8). The inevitable conclusion is that water supply is not a priority in Tonga.

This is in contrast to the village community development plans were 55% of villages ranked water supply as the number 1 priority and 91% of villages ranked water supply as a priority concern.

  1. Explain the case of the Kingdom of Tonga situation

More details concerning the Kingdom of Tonga are now provided in section 2.1.

  1. Show results for the applied methodology

The results of our methodology are presented in Tables 1 to 6 and 8 to 9, and Figures 2 to 6. Table 8 is new showing the infrastructure services analysis in TSDFII.

  1. Results and discussion (also analyze the applied methodology)

We have now included Section 4.3 Limitations of this work which provides a critical analysis of the applied methodology.

  1. Conclusion (summarize results and discussion, what is new has been learned/concluded, how can the presented methodology be used in similar cases, how can this improve planning (especially in water (supply) management in other SIDS) what are the recommendations

These are excellent points. We have completely rewritten the Conclusion to address these points specifically.

Specific Comments

 The manuscript presents the analyses of the GAP between national and local sustainable development planning for water supply in the Kingdom of Tonga.

Mentioning SDG6 in the title should be omitted because only a minor part of the manuscript refers to it (few sentences in Tittle, Introduction, 3.8. in Results and  4.2. in Discussion).

Now more emphasis has been placed on SDG6. TSDFII claims that it fulfills Tonga’s commitments to SDG6. There is no emphasis on water and no mention of sanitation. NSDS are claimed to be valuable because they not only meet the country’s needs, but they fulfil obligations to SDGs – here we show that that is not always true.

Although the collection of data and the analyses are presented well, and are very interesting, there is no additional new knowledge or methodology developed. How can the facts derived from analyses and presented in conclusion improve the water supply in Tonga or in similar countries?

We disagree with this comment. The new knowledge presented here are:

  • The relation of medium rainfall trends in Tonga to the medium trends in the PDO. It has been widely claimed that trends over the last 30 years in rainfall are due to climate change, here we show that this is not so in Tonga the trends in rainfall over 31-year periods are due to trends in PDO not climate change.
  • Improved governance is one of the keys to improved water management, particularly in Pacific Island Countries. NSDS have been widely promulgated by the UNDP, World bank and ADB as the principle instrument for improving national governance. They determine Government effort and the flow of resources for a decade. Here we show that the TSDFII has totally failed to reflect the local village community priorities across the whole country for improved water supply. This is a major governance failure that has not been demonstrated previously. It has fundamental importance to how NSDS are developed in SIDs.

Scientific papers should present new knowledge, methods, methodologies that should help improving (in this case) the water management and sustainable development planning. The paper written in this way has no scientific added value to water management.

We disagree with this comment but admit that we did not make our contributions clear enough. The failures of TSDFII are due to the fact that it did not use Census and available hydrological data and did not pay any attention to local community concerns. Since NSDS determine the national flow of resources, these means that water supply is not adequately resourced and needs attention. With the results of this work we were able to convince the Government of Tonga to pass the Tonga Water Resources Act 2020. Prior to that the water resources of Tonga had no legal protection from contamination, overuse and no imperative for conservation. There is a wider problem here, NSDS have been used throughout the Pacific region where safe and adequate water supply is a matter of life and death. How many others have not paid attention to local needs?   

In the Introduction it is important to analyze the existing knowledge, methods and methodologies regarding the topic of the article:

  • Do similar problems of differences between national and local planning regarding water supply exists in other SIDSs? Are there any articles that cover this problem or the problem of prioritizing the water supply?

That is a research question which we are currently pursuing. Tonga presented a unique opportunity with both national as well as local village development plans available.

  • How did you define the methodology that was used in this manuscript? Are there other methodologies that can be used for analyzing the differences between national and local planning (literature overview).

The methodology we have used relies on defining differences for one particularly sector, water supply. We have chosen water supply because adequate and safe water supply is basis for almost all the other SDGs. We simply search the main planned outcomes of TSDFII and some of the concepts raised in discussion for water, water supply, WaSH etc. Following the reviewer’s comments, we have compared them with the weight given to other infrastructure services. As we explain in the methodology, for CDPs we search the ranked priorities given by each village for water, water supply, WaSH etc and determine the percentage of villages that rank water supply top priority, within the top 3 priorities or list it as a priority for each Island Division.   

I suggest to rewrite the paper in the way to stress on:

  • The methodology that has been used for the analyses (present on flowchart, explain in text)
  • How it can help in defining the GAP on different levels of planning especially for SIDS?
  • What are the recommendations for improving the water management and sustainable development planning (lessons learned)?
  • What are the suggestions to improve the water supply in the Kingdom of Tonga, based on identified GAP?

We have substantially rewritten the paper taking into account these points which has improved the paper substantially.

Reviewer 3 Report

The manuscript is clear, well structured, with a good scientific soundness and can be published in its current form. No further comments are needed. I just ask you to make some changes in the Abstract: 1) reduce the details on the study case; 2) generalize the issue and for this reason it would be enough to report the lines 625-627 in the abstract.

Author Response

Responses to Review No 3

The manuscript is clear, well structured, with a good scientific soundness and can be published in its current form. No further comments are needed. I just ask you to make some changes in the Abstract: 1) reduce the details on the study case; 2) generalize the issue and for this reason it would be enough to report the lines 625-627 in the abstract.

Thank you. The Abstract has now been revised as suggested

Round 2

Reviewer 2 Report

Dear authors, 

the structure of the article with 3 questions in the Introduction and the analyses done in order to answer them significantly improved the article. 

Now the novelty/originality is explained better, and the methodology applied too.

Since most comments were implemented or explained the article can be accepted.

Kind regards